# Genomic and Epigenetic Changes Drive Aberrant Skeletal Muscle Differentiation in Rhabdomyosarcoma

**DOI:** 10.3390/cancers15102823

**Published:** 2023-05-18

**Authors:** Silvia Pomella, Sara G. Danielli, Rita Alaggio, Willemijn B. Breunis, Ebrahem Hamed, Joanna Selfe, Marco Wachtel, Zoe S. Walters, Beat W. Schäfer, Rossella Rota, Janet M. Shipley, Simone Hettmer

**Affiliations:** 1Department of Hematology/Oncology, Cell and Gene Therapy, Bambino Gesù Children’s Hospital, IRCCS Istituto Ospedale Pediatrico Bambino Gesu, Viale San Paolo 15, 00146 Rome, Italy; 2Department of Clinical Sciences and Translational Medicine, University of Rome Tor Vergata, Via Montpellier 1, 00133 Rome, Italy; 3Department of Oncology and Children’s Research Center, University Children’s Hospital of Zurich, 8032 Zürich, Switzerland; 4Department of Pathology, Cell and Gene Therapy, Bambino Gesù Children’s Hospital, IRCCS, Viale San Paolo 15, 00146 Rome, Italy; 5Division of Pediatric Hematology and Oncology, Department of Pediatric and Adolescent Medicine, University Medical Center Freiburg, University of Freiburg, 79106 Freiburg, Germany; 6Sarcoma Molecular Pathology Team, Divisions of Molecular Pathology and Cancer Therapeutics, The Institute of Cancer Research, London SM2 FNG, UK; 7Translational Epigenomics Team, Cancer Sciences, Faculty of Medicine, University of Southampton, Southampton SO17 1BJ, UK; 8Spemann Graduate School of Biology and Medicine (SGBM), 79104 Freiburg, Germany; 9Comprehensive Cancer Centre Freiburg (CCCF), University Medical Center Freiburg, 790106 Freiburg, Germany

**Keywords:** rhabdomyosarcoma, skeletal muscle, differentiation, genomics, epigenetics

## Abstract

**Simple Summary:**

Rhabdomyosarcoma is the most common soft tissue cancer in children and adolescents. Its resemblance to skeletal muscle tissue distinguishes rhabdomyosarcomas from other types of soft tissue cancer. The development and integrity of healthy skeletal muscle depend on a strictly regulated, hierarchically organized machinery in cells. In rhabdomyosarcoma, this process goes awry, resulting in aberrant, malignant skeletal muscle states. These aberrant skeletal muscle states define subtypes of rhabdomyosarcomas. In this review, we describe normal muscle development and summarize recent insights into how changes in rhabdomyosarcoma cells disrupt normal skeletal muscle homeostasis, thereby defining the cancerous nature of this disease. We also describe differences in myogenic differentiation characteristics between different groups of cells within the tumors. Such differences appear to be dynamic and influence the behavior of the cells. We believe that interactions between cancer genes/proteins and basic muscle programs are key to understanding the cancerous identity of rhabdomyosarcoma and may provide windows of opportunity with regard to rhabdomyosarcoma treatment.

**Abstract:**

Rhabdomyosarcoma (RMS), the most common soft-tissue sarcoma in children and adolescents, represents an aberrant form of skeletal muscle differentiation. Both skeletal muscle development, as well as regeneration of adult skeletal muscle are governed by members of the myogenic family of regulatory transcription factors (MRFs), which are deployed in a highly controlled, multi-step, bidirectional process. Many aspects of this complex process are deregulated in RMS and contribute to tumorigenesis. Interconnected loops of super-enhancers, called core regulatory circuitries (CRCs), define aberrant muscle differentiation in RMS cells. The transcriptional regulation of MRF expression/activity takes a central role in the CRCs active in skeletal muscle and RMS. In PAX3::FOXO1 fusion-positive (PF+) RMS, CRCs maintain expression of the disease-driving fusion oncogene. Recent single-cell studies have revealed hierarchically organized subsets of cells within the RMS cell pool, which recapitulate developmental myogenesis and appear to drive malignancy. There is a large interest in exploiting the causes of aberrant muscle development in RMS to allow for terminal differentiation as a therapeutic strategy, for example, by interrupting MEK/ERK signaling or by interfering with the epigenetic machinery controlling CRCs. In this review, we provide an overview of the genetic and epigenetic framework of abnormal muscle differentiation in RMS, as it provides insights into fundamental mechanisms of RMS malignancy, its remarkable phenotypic diversity and, ultimately, opportunities for therapeutic intervention.

## 1. Introduction

Rhabdomyosarcoma (RMS), the most common soft-tissue sarcoma in children and adolescents, comprises a diverse group of cancers that are distinguished from other types of soft-tissue sarcomas by the presence of skeletal muscle features [1,2].

Normal skeletal muscle development combines the sequential expression of muscle-specific transcription factors (TFs) and terminal withdrawal from the cell cycle to produce a rather large organ, which accounts for 38% of total body mass in men and 30% in women. Skeletal muscle is composed of bundles of terminally differentiated myofibers, which contain multiple post-mitotic nuclei alongside myofibrils consisting of sarcomeres in repeating series. Two main protein filaments, actin and myosin, slide past each other and thereby change sarcomere length. This produces the vital forces necessary to breathe, maintain posture and move. Movements influence the production of cytokines, small peptides, and proteoglycans, which regulate muscle growth, differentiation, and remodeling [3]. Throughout life, muscle integrity is maintained by a population of muscle-resident stem cells, called satellite cells, which are positioned between the basal lamina and the plasmalemma of myofibers. Upon activation, a group of highly homologous transcription factors—called myogenic regulatory factors (MRFs)—govern satellite cell activation, myoblast proliferation and fusion into new myofibers, all the while maintaining the pool of satellite cells required to maintain muscle integrity in the future [4].

RMS partially recapitulates skeletal muscle development, but terminal skeletal muscle differentiation is blocked in RMS cells [5,6,7,8]. In this review, we provide an overview of recent insights into the genetic and epigenetic landscape of RMS and how these advances may shape our understanding of disease-driving mechanisms in RMS. As skeletal muscle features define RMS identity and distinguish the disease from other types of soft-tissue sarcomas [1,2], understanding the cellular and molecular underpinnings of aberrant skeletal muscle differentiation and how RMS-relevant oncogenic alterations hijack the complex processes governing muscle differentiation provides fundamental insights into heterogeneity across the RMS spectrum and may provide opportunities for therapeutic intervention.

## 2. Normal Skeletal Muscle Development and Homeostasis

### 2.1. The Myogenic Regulatory Factor (MRF) Family

The myogenic regulatory factors (MRFs) comprise Myogenic Differentiation 1 (*MYOD1*) [9], Myogenin (*MYOG*/*MYF4*) [10], Myogenic factor 5 (*MYF5*) [11], and Myogenic factor 6 (*MRF4*/*MYF6*) [12]. MRFs likely evolved from a common ancestral gene and retained a certain degree of functional overlap. They are considered master regulators of myogenesis. Their exogenous expression induces lineage reprogramming and drives non-myogenic cells toward a skeletal muscle fate [13,14]. In healthy adult muscle, transcript levels of *MYF6* are the highest of all MRFs, while *MYOD1* and *MYF4* transcript levels are generally low. *MYF5* is expressed in quiescent satellite cells [15].

We note that skeletal muscle contains two types of muscle fibers: slow twitch (type I) and fast twitch (type II). Most muscles are composed of a mixture of slow-twitch and fast-twitch fibers. Slow twitch myofibers—making up most of the soleus muscle and the back muscles—contain Myosin heavy chain (MyHC) isoforms with lower ATPase activity and provide skeletal support/resistance to fatigue. Fast twitch myofibers—a large portion of the small muscles in hands and eyes—contain MyHC isoforms with higher ATPase activity and utilize glycolysis to rapidly produce ATP and support quick, high-power movements. *MYOD1* is preferentially expressed in fast-twitch myofibers and *MYF4* in slow-twitch myofibers [4].

During development and regeneration, MRFs collaborate in directing progenitor cells to establish the skeletal muscle lineage. *MYOD1* and *MYF5* are generally viewed as factors responsible for the myogenic determination of cells, while *MYF4* and *MYF6* are linked to terminal differentiation [13,16].

### 2.2. Embryonal Skeletal Muscle Development

During embryonal development, skeletal muscles of the head originate from unsegmented cranial mesoderm. Skeletal muscles of the trunk and limb derive from paraxial mesoderm, which segments into pairs of somites, whose dorsolateral parts form the dermomyotome. The hypaxial lip of the dermomyotome provides PAX3+/MRF− cells, which migrate to form a portion of the tongue, some neck/jaw muscles, and all limb muscles; they express MRFs post-migration. By contrast, the dermomyotome supplies MYF5+/MYF6+ cells to the early myotome; these cells are depleted later on and replaced by PAX3+/PAX7+ cells to form the segmented muscles associated with vertebrae and rips. Ultimately, MYF4 governs the myogenic differentiation program along with MYOD1 and MYF6 [4,17,18,19,20]. Gene inactivation studies in mice have shown that viable animals can be generated after the inactivation of MYF5, MYF6, or MYOD1, although the combined loss of MYF5, MYF6, and MYOD1 results in a complete failure of the myogenic specification [21]. However, MYF4 was shown to be indispensable for myogenic differentiation, and MYF4-null mice die at birth [22].

### 2.3. Post-Natal Regeneration of Skeletal Muscle

Throughout life, there is enormous regenerative potential in skeletal muscle. The growth and repair of skeletal muscle after birth depend on satellite cells expressing PAX7 and, in certain muscles, PAX3. In adult mice, in which the *Pax-3* gene was targeted with *nLacZ* reporters, evidence for Pax-3 expression in satellite cells was found in the diaphragm, in 50% of forelimb muscles, in the gracilis muscle, in the ventral trunk and in body wall muscles, but not in intercostal muscles, in muscles of the head and in most hindlimb muscles [23]. Still, PAX7 is considered the master regulator of satellite cell function [24]. PAX3 cannot replace the anti-apoptotic functions of PAX7, resulting in progressive postnatal loss of satellite cells in *Pax7*-deficient, *Pax3*-expressing mouse muscles [23].

Developmentally, satellite cells derive from the same founder cells that form the muscles they reside in; *PAX3* and *PAX7* appear to be redundant during this process [23]. *MYF5*, *MYOD1*, and *MYF6* were all shown to be active at some point during satellite cell specification, followed by the down-regulation of *MYOD1* and *MYF6* during further development [4]. During muscle regeneration post-injury (Figure 1), MRFs are redeployed to activate satellite cells and govern myoblast proliferation/fusion.

*MYF5* is the only active MRF expressed alongside *PAX7* in quiescent satellite cells in adult skeletal muscle. Injury rapidly induces *MYOD1* expression. *MYOD1*-/*PAX7*-/*MYF5*-myoblasts continue to proliferate and down-regulate *PAX7* while maintaining *MYOD1*, before they finally commit to myogenic differentiation via induction of *MYF4*. Other myoblasts maintain *PAX7* but down-regulate *MYOD1*, and ultimately withdraw from the cell-cycle to regain quiescence and repopulate the satellite cell pool [25,26,27]. We note that the sequential activation of MRFs in satellite cells appears to be bidirectional both during developmental specification and muscle regeneration [4].

### 2.4. Epigenetic Regulation of Muscle Development/Differentiation

Generally, MRFs act as tissue-restricted transcription factors to remodel chromatin structure (Figure 1) and allow for the binding of the transcriptional machinery at muscle-specific genes. MRF function depends on heterodimerization with members of the E-protein family of basic helix-loop-helix (bHLH) proteins, which are ubiquitously expressed [13,28]. The MRF/E-protein heterodimeric complexes bind the E-box consensus sequence (CANNTG) in the regulatory regions of muscle-specific genes. This interaction is limited by chromatin accessibility. While MYOD1 binds many genes in muscle cells, it modifies gene expression only at a fraction of its targets. Instead, MYOD1 binding correlates with the opening of the chromatin structure at target genes through the acetylation of histones, thereby permitting MYF4 binding [13].

Mitogen-Activated Protein Kinase 14 (p38α/MAPK14, hereafter referred to as p38α) is a critical regulator of muscle differentiation. Chromatin binding of p38α to a large set of active promoters drives the expression of muscle differentiation-relevant transcripts. Forced activation of p38α in myoblasts results in terminal muscle differentiation [29]. Also, expression of muscle-specific TFs is controlled by specific actions of various chromatin modifiers such as mammalian SWItch/Sucrose Non-Fermentable (mSWI/SNF) and Polycomb repressive complex 2 (PRC2). When satellite cells are induced to differentiate, the PRC2 component Enhancer of Zeste 2 Polycomb Repressive Complex 2 Subunit (EZH2) is phosphorylated by p38α, which leads to its recruitment to the *PAX7* locus and, subsequently, repression of PAX7 expression. In turn, when myoblasts initiate differentiation, EZH2 departs from promoters of muscle-specific genes to allow their expression [30].

## 3. Genetic and Epigenetic Landscape of RMS

### 3.1. RMS Classification

The World Health Organization recognizes four RMS histotypes, i.e., embryonal, alveolar, spindle-cell sclerosing and pleomorphic RMS [31], although true phenotypic heterogeneity across the RMS spectrum goes far beyond this classification system [1]. The two main histological subtypes diagnosed in the pediatric population are alveolar and embryonal RMS [1,2]. Tumors with pleomorphic and not otherwise specified (NOS) histology account for the majority of RMS diagnosed in individuals >18 years of age [31,32].

While RMS histological criteria have changed over time [2], the presence of the *PAX3::FOXO1* (P3F) fusion gene has been associated with significant negative prognostic value [33,34,35,36] and distinct tumor biology [37,38,39]. These observations have led physicians and scientists to consider *PAX::FOXO1* fusion-positive (PF+) and *PAX::FOXO1* fusion-negative (PF−) RMS as profoundly different disease categories. The P3F fusion oncogene more or less exclusively occurs in alveolar histology RMS. Nevertheless, we strongly caution against equating PF+ and alveolar histology RMS. This simplified view fails not only alveolar histology tumors that do not express PF (their biology continues to be an area of active investigation), but also the distinct oncogenic impact of *PAX7::FOXO1* (P7F, another area of active investigation). For the purpose of this overview of the genetic landscape of RMS, we distinguish alveolar RMS (including PF+ and PF− alveolar histology tumors) and PF−, non-alveolar RMS.

### 3.2. Genetic Landscape of Alveolar RMS

Alveolar RMS has long been associated with very poor survival [40,41]. In approximately 80% of alveolar histology cases [31,42,43], balanced t(2;13)(q35;q14) and t(1;13)(p36;q14) translocations [44,45] result in expression of fusion oncoproteins consisting of the N-terminal DNA-binding domains of *PAX3* or *PAX7* and the C-terminal transcription activation domain of *FOXO1* [46,47]. Retrospective studies suggest that the *FOXO1* translocation partner may hold prognostic significance with superior outcomes for P7F+ compared to P3F+ tumors [34,35,43,48,49,50]. Generally, *PAX::FOXO1* fusion-positive (PF+) alveolar histology tumors are characterized by extremely low sequence variation rates [38,51], distinct DNA methylation patterns [39,52,53,54] and frequent amplification events [51,55,56]. Classic regions of chromosomal amplification include the 12q13-14 amplicon containing *CDK4* [51,55] and the 2p24 amplicon containing the *MYCN* oncogene [55,56].

Among *PAX3/7::FOXO1* fusion-negative (PF−) alveolar histology RMS, occasional alternative *PAX* gene translocations (e.g., *PAX3::AFX* (*FOXO4*), *PAX3::NCOA1*, *PAX3::NCOA2* and *PAX3::INO80D* [48,57,58,59]) have been identified and may be predictive of aggressive behavior. For the remaining *PAX::FOXO1* fusion-negative (PF−) alveolar histology tumors, gene expression profiling revealed profound overlap with embryonal histology tumors, which has brought forth recommendations to reduce the intensity of therapy for patients with PF−, localized alveolar histology tumors [34,60].

### 3.3. Genetic Landscape of PF−, Non-Alveolar RMS

In children and adolescents, PF− RMS typically exhibits embryonal histology and more favorable outcomes. Whole genome, whole exome and hybrid capture panel sequencing demonstrated that the frequency of mutations in PF− RMS was consistently higher than in PF+ RMS [38,51,53,61,62,63]. The most common finding in PF− tumors was a mutation of at least one member of the RAS pathway (detected in more than 50% of all cases; [38]).

Approximately 30% of PF− RMS exhibit non-satisfactory responses to treatment and poor survival. *TP53* variants, detected in 12–13% of PF− cases [38,51], correlated with inferior outcomes in PF− RMS (51,53,61). Besides, somatic variants in exon 1 of the *MYOD1* gene (c.365G > Tp.L122R) were associated with aggressive behavior and poor outcomes among PF− RMS. Specifically, *MYOD1* mutations at the hotspot codon L122R were observed in 3% of all PF− RMS cases and often co-occurred with other gene mutations, most notably variants in *PIK3C*, RAS pathway genes or *CDKN2A* [51,64].

Of note, somatic L122R mutations in the *MYOD1* gene were first detected in older patients diagnosed with RMS with spindle cell/sclerosing morphology and extremely aggressive clinical course [64,65,66]. Yet, spindle cell RMS may also be diagnosed in very young children under one year of age. These prognostically rather favorable RMS manifestations [67] have been associated with non-*PAX* gene translocations, where the genes encoding the Serum Response Factor (*SRF*) and the TF Vestigial Like Family Member 2 (*VGLL2*) fuse with the Nuclear Receptor Coactivator (*NCOA2;* [66,68]) or Cbp/P300 Interacting Transactivator With Glu/Asp Rich Carboxy-Terminal Domain 2 (*CITED2*; [66]).

### 3.4. Methylation Profiling of RMS

Methylation profiling appears to recapitulate some of the previously described histological and genetic heterogeneity of RMS. Clustering based on differences in DNA methylation distinguishes between embryonal and alveolar RMS [39,52,53,54] and identifies subsets of tumors corresponding to adult-type pleomorphic RMS or spindle-cell sclerosing histology RMS with *MYOD1* p.L122R variants [52]. DNA methylation also correlated with the presence of *DICER1* variants in tumor tissue [69] and appeared to predict outcomes among embryonal RMS [53]. Methylation profiling may thus aid in recognition of biologically and clinically distinct RMS manifestations.

### 3.5. Imprinting in RMS

Imprinting—an epigenetic mechanism whereby gene expression is dictated by germline expression of the maternal (or paternal) allele through differences in methylation at “imprinting control region” (ICRs) rich in cytidine-phosphate-guanosine (CpG) repeats [70]—appears to contribute to RMS malignancy. In many embryonal histology RMS tumors, the D11S988 locus on chromosome 11p15.5 was identified as a common region of loss of heterozygosity [71] and disruption of imprinted genes, including Insulin-like Growth Factor 2 (*IGF2*), H19 Imprinted Maternally Expressed Transcript (*H19*), and Cyclin-Dependent Kinase Inhibitor 1C (*CDKN1C*; [72]). The 11p15.5 locus contains two imprinting centers, IGF2/H19 and CDKN1C/KCNQ1OT1 and was found to be aberrantly methylated in embryonal and alveolar RMS [73].

### 3.6. Genetic RMS Susceptibility due to Pathogenic Germline Variants in Cancer Genes

It has long been established that germline variation confers genetic RMS susceptibility [74]. Pathogenic/likely pathogenic (P/LP) germline variants in cancer predisposition genes were detected in 7–17% of children, adolescents and young adults with RMS who were included in published germline sequencing cohorts [75,76,77,78,79]. Specific RMS phenotypes [80] were linked to P/LP variants in certain cancer predisposition genes, such as P/LP *TP53* germline variants (anaplastic histology, [81]) and P/LP *DICER1* germline variants (location of primary tumors in the female urogenital tract, [82]). The most frequent P/LP germline variants associated with young-onset RMS were *TP53*, *NF1*, and *BRCA2* [75,76,77,78,79]. We note that childhood cancer risk in *BRCA1/2* germline variant carriers has been deemed negligible. In-depth studies, including parallel tumor/germline sequencing, are necessary to further explore the possible causal relationship between the development of RMS (including associated phenotypes and disease risk) and *BRCA2* and other germline variants [80].

## 4. Cell Cycle Progression in Skeletal Muscle and RMS

### 4.1. Cell Cycle Regulation in Skeletal Muscle

To maintain the regenerative potential of skeletal muscle, cell cycle regulation must allow for satellite cell quiescence, the rapid proliferation of myoblasts, and orderly exit from cell division of terminally differentiating myocytes. MYOD1/MYF5 and MYF4 have important and opposing roles in the regulation of this process.

Cell cycle progression results from consecutive activation and inhibition of phosphoproteins by cyclin-dependent kinases (CDKs), which are bound by activatory cyclins. These complexes are regulated by cyclin-dependent kinase inhibitors (CDKIs). MYOD1 and MYF5 drive the expression of transcripts with critical functions in the cell cycle to promote the expansion of the muscle satellite cell pool/myoblast proliferation [83,84]. The functions of MYOD1 and MYF5 are non-redundant. Importantly, MYOD1 (not MYF5) plays an important role in mediating DNA damage response. Phosphorylation of MYOD1 at Tyr30 is inhibited by c-Abl tyrosine kinase in response to genotoxic stress [85]. This prevents myotube formation in cells with DNA damage [13].

In order to complete myogenic differentiation, myoblasts must exit the cell cycle (Figure 1). MYOD1 and MYF4 act in a coordinated fashion to orchestrate cell cycle exit during muscle differentiation. While MYOD1 plays an important role and *MYOD1*−/− cells fail to exit the cell cycle, MYF4 assumes the key role in this process. Ectopic expression of *MYF4* mediates cell cycle exit by activating key genes (e.g., retinoblastoma susceptibility protein (*RB*), *CDKN1A*, miR17-92 cluster of microRNAs) involved in suppressing translation and transactivation of E2F transcription factors (*E2F*) [13]. When *RB* and Cyclin-Dependent Kinase Inhibitor 2A (*CDKN2A*) are downregulated, *MYF4*-expressing myocytes may re-enter the cell cycle [86].

The transcriptional activity and protein availability of MRFs during cell cycle progression are modulated through phosphorylation-dependent degradation by the ubiquitin-proteasome system or inhibition of their DNA binding/transcriptional activation ability [13]. Myf5 protein levels peak in G0, decline during G1 and rise again at the end of G1. MYOD1 protein levels peak in mid-G1, are downregulated during G1/S transition (important as MYOD1 blocks G1/S transition), and rise again at G2/M. MYF5 and MYOD1 levels are similar at the transition from G2 to M (Figure 1) [87,88].

### 4.2. Cell Cycle Regulation in RMS

In RMS, Stewart et al. employed integrated genomic, epigenomic and proteomic analyses to pinpoint activation of the cyclin D/CDK4/6-RB/E2F pathway and alteration of the G2/M mitotic spindle checkpoint pathway in RMS cells [89].

Further evidence indicates that PF might be involved in cell cycle regulation. In cells derived from a conditional genetic mouse model of P3F+ RMS, dynamic expression of P3F at the single-cell level was observed [90]. Interestingly, the P3F^high^ subset of mouse RMS cells contained a higher proportion of cells in G2/M than P3F^low^ expressing cells, which were mostly in G0/G1 [90,91]. High levels of P3F in G2 appeared to facilitate the transition to M, thereby mediating checkpoint adaptation under stress conditions and tolerizing RMS cells to irradiation/chemotherapy [90].

### 4.3. Aneuploidy in RMS

Activation of two P3F alleles drastically enhanced tumor development in the mouse model of P3F+ RMS [92]. Interestingly, in human alveolar RMS, PF expression is upregulated by different mechanisms. Duplication is the second most frequent event after PF translocation. P7F is often amplified. This has been recognized early on with relatively simple studies of DNA content establishing that alveolar histology was strongly associated with near tetraploidy (1.80 to 2.60 times the DNA content of normal cells; [93]). Sequencing studies recently confirmed this observation [38].

## 5. Developmental Myogenic Heterogeneity of RMS

### 5.1. Cell-to-Cell Heterogeneity within the Cancer Cell Pool

Tumors are complex systems with single cells as their fundamental unit. The characterization and classification of tumor cells have puzzled cancer biologists for a long time, but traditionally relied on microscopic observations and protein staining techniques, which are low-throughput, and therefore offer a limited and biased understanding of intratumoral heterogeneity. Recently, single-cell RNA sequencing (scRNAseq) has emerged as a powerful tool to measure gene expression levels from thousands of individual single cells. ScRNAseq offers an unbiased picture of cancer cell subpopulations and thereby has the potential to revolutionize our understanding of cancer [94].

### 5.2. Aberrant MRF Expression in RMS Tissue

Immunohistochemical staining for myofilaments and MRFs has been used to facilitate diagnosis and distinguish RMS from its histological mimics and pinpoint aberrant myogenic differentiation states across the diverse spectrum of this cancer. While *MYOD1* and *MYF4* expression is very low in adult skeletal muscle [4], most RMS tumors express easily detectable amounts of MRFs [95]. In embryonal histology RMS, MYF4-positive tumor cells are typically interspersed with many MYF4-negative cells. Alveolar histology RMS has been associated with strong expression of *MYF4* in virtually all nuclei [96], which may indicate a late block in myogenesis (after expression of MYF4) or oncogenic processes resulting in strong expression of *MYF4*. Spindle cell/sclerosing RMS typically exhibits weak to absent MYF4 but strong nuclear MYOD1 staining [2].

### 5.3. Developmental Heterogeneity at the Single Cell Level within the RMS Cell Pool

Elegant studies employing scRNAseq to investigate the transcriptional heterogeneity within the RMS cell pool revealed that both alveolar and embryonal RMS harbor cells with myogenic potential, which are stalled at distinct developmental states. These include a transcriptionally immature stem cell-like/mesoderm/mesenchymal state (expressing *PAX3*/*PAX7*/*CD44*), a myoblast state (expressing *MYF5*/Cell Division Cycle 20 (*CDC20*)/Cyclin B1 (*CCNB1*)) containing highly proliferative/cycling cells, and a more differentiated myocyte state (expressing *MYF4*/*MYH3*/*MYH8*) associated with better patient outcomes (Figure 2) [97,98,99]. It is important to note that the tumor cell pool spanned a wide range of developmental indices in embryonal RMS, while it was more skewed towards later stages of myogenesis in alveolar RMS [97,98,99], which is consistent with previously published immunohistochemical findings described above [82].

PF+ and PF− RMS appear to be arrested at specific stages of skeletal muscle development. The core transcriptional signature of PF+ RMS correlates with the signature found in developing muscle at 7–7.75 weeks of age, at which time embryonic muscle transitions to fetal muscle [99]. By contrast, the PF− RMS core signature was expressed in both embryonic and fetal muscle cells, but not in adult muscle [99].

### 5.4. Cellular Hierarchies in RMS

RNA velocity analysis and/or lentiviral barcoding of cells within patient-derived xenografts (PDX) have been applied to show that RMS tumors are hierarchically organized and recapitulate normal myogenesis, with cells transitioning from the undifferentiated stem-like/mesoderm state through the myoblast to the differentiated myocyte state [97,98]. However, recent observations suggest substantial differences in the organization of cellular hierarchies across the RMS spectrum, with PF− RMS following a possible, stem-like cell-driven model and prominent plasticity of PF+ RMS.

Wei et al. used lineage barcoding and functional assays to show that prospectively isolated CD44+/CD90+ or CD90+/CHODL+ cells of the mesenchymal subpopulation establish more, larger tumorspheres and exhibit more efficient, faster tumor growth in mice [99]. Previous studies using fluorescent transgenic zebrafish bearing KRAS(G12D)-induced RMS yielded similar insights [100]: GFP-labeled MYF5^high^ zebrafish tumor cells (expressing markers of satellite and early muscle progenitor cells) and mCherry-labeled MYF4^high^ zebrafish tumor cells (expressing high levels of mature muscle markers) were isolated by fluorescence-activated cell sorting and transplanted. Tumor-propagating capacity was enriched within the MYF5^high^ cell population, while MYF4^high^ cells were prone to migrate, invade through a normally impenetrable collagen matrix and enter the vasculature. In zebrafish, slow-moving MYF5^high^ cells were found in newly colonized regions after initial invasion by MYF4^high^ cells [100]. These studies indicate that the mesenchymal state characterizes the only tumor-propagating subpopulation—i.e., a stem-like population of cells capable of re-creating original tumor heterogeneity—in PF− RMS [99].

By contrast, PF+ RMS displays more prominent intrinsic plasticity, with both stem-like and cycling-myoblast states having the capacity to re-create the original composition of the tumor. These observations are consistent with in vivo tumor-propagating capacity of all single-cell clone-derived clones of P3F+ RMS cells examined by Generali et al. [101]. Of note, fluctuating levels of P3F expression in single, mouse P3F+ RMS cells were linked to differences in in vitro clonal activity and in vivo tumor-propagating capacity, indicating plasticity in P3F expression at the single-cell level [91]. Similar to what has been proposed for certain other cancers [102], cellular plasticity may be the mechanism at the root of PF+ RMS aggressiveness [97].

### 5.5. Differential Drug Responsiveness of RMS Cell Subsets

The functional consequences of intratumoral heterogeneity on drug response are an emerging topic of interest. Lentiviral barcoding of cells within PDX tumors followed by scRNAseq revealed that the highly proliferative, myoblast-like population of PF− RMS cells was more sensitive to conventional chemotherapy, while the more quiescent, mesoderm-like population was more resistant (Figure 2). Following chemotherapy, there was a decrease in clonal diversity, and the quiescent population of immature mesoderm-like cells expanded to reconstitute the developmental hierarchy of the tumor [98].

A similar phenomenon was observed in PF+ RMS, where—following in vitro treatment with etoposide or vincristine—the cycling/myoblast state was depleted, and stem-like cells were enriched [97]. This shift towards stem-like cell states could be explained by two scenarios: (i) selection of more resistant cells similar to what has been observed in PF− RMS, (ii) cells in the cycling/myoblast state and/or more differentiated cells may transition to stem-like states in response to treatment, or (iii) stem-like cells may increase their proliferation under therapy pressure, as it was recently indicated in glioblastomas [102]. Taken together, it is conceivable that chemotherapy drives a shift in RMS cellular states toward stem-like states.

These findings suggest that subpopulations of cells within the RMS cell pool are locked in distinct cell states, including an immature stem-cell state and more mature muscle states. Differentiated cells represent a minority of the tumor, do not cycle, have lower malignant potential and correlate with better patient outcomes (Figure 2). Similar developmental hierarchies (with subpopulations of stem-like cells capable of self-renewal and more differentiated cells with lower capacity to initiate tumors) have been identified in other pediatric tumor types, such as glioma [103], neuroblastoma [104], medulloblastoma [105], and rhabdoid tumors [106]. Shifting undesired stem-like states into “desired” differentiated states could provide therapeutic benefits in such cancers. This will require an in-depth understanding of how transitions between RMS cell states are regulated and how these processes can be manipulated therapeutically.

## 6. Aberrant Myogenic Differentiation due to Miswiring of Core Regulatory Circuits (CRCs) in RMS

### 6.1. Regulation of Cell Fate by CRCs

The molecular mechanisms involved in blocking differentiation in RMS are an area of active research. In general, cell type-specific transcriptional programs that govern cell fate specificity are tightly controlled by a highly coordinated array of master TFs (MTFs). These MTFs are organized in interconnected transcriptional loops, called core regulatory circuitries (CRCs).

Within CRCs, individual MTFs control their own expression as well as the expression of the other involved TFs and exert feed-forward transcriptional control [107]. These transcriptional networks ultimately control key transcriptional programs and drive processes such as myogenic differentiation in the muscle lineage. At the chromatin level, MTFs are accumulated at super-enhancers (SEs) and function under the control of SEs (Figure 3). Hence, insights into the SE landscape and associated TF binding motifs can be exploited bioinformatically to predict the CRCs that are active in individual cell types (Figure 3).

Cell-type specific CRCs have been identified for a number of different tissues, including normal skeletal muscle. Two families of TFs are at the center of the CRC active in skeletal muscle cells: the MRF and the myocyte enhancer factor 2 (MEF2) family [108,109].

### 6.2. Reshaping of Skeletal Muscle CRCs in RMS

The normal muscle CRC is active in both PF+ and PF− RMS. However, in both cases, the muscle CRC is further complemented with additional modules [5], including a pan-RMS module composed of MYOD1 and MYF4 and additional subtype-specific modules (Figure 3).

The P3F+ RMS-specific module relies on P3F and MYCN [5,110,111,112]. P3F was shown to have pioneer factor activity and shape the SE landscape in PF+ RMS cells, thereby demonstrating its importance for cell fate regulation [5,113]. Importantly, the expression of P3F is under the control of a FOXO1 enhancer that is normally active in late-stage myogenesis (Figure 3A) [6]. This P3F+ RMS-specific module replaces the originally sequentially organized CRC active in muscle precursor cells by an infinite loop that blocks the cells in an undifferentiated state.

By contrast, the PF− RMS-specific module involves *PAX7* and Activator Protein-1 (AP1)-family TFs (Figure 3B) [5]. Additional TFs were found to be involved in CRCs in specific RMS cell lines (dbCoRC database http://dbcorc.cam-su.org/ accessed on 8 December 2022).

### 6.3. Differential Use of CRCs at the Single Cell Level

We note that the studies defining the CRC landscape in RMS were performed with bulk cell populations. As described above, recent scRNAseq and ATACseq analyses identified different subpopulations of cells to be present in RMS tumors [97,98,99]. These subpopulations of cells are aligned along the myogenic lineage, from muscle stem cell-like cell states to myosin-heavy chain positive, differentiated cell states. This has been linked to the differential use of CRC SEs [98], but the exact CRCs involved in the different cell states are yet to be determined.

It remains to be seen if some cells may be able to escape, potentially due to the influence of the microenvironment. Strong differentiation tendencies have been observed in RMS cells when they were removed from their regular in vivo niche and transferred into culture dishes [114]. Changes in nutrient availability, surface matrices, proximity to other tumor cells, etc., may also trigger the transition between different RMS cell states [91].

### 6.4. Evolving Understanding of Tumor Dependency Concepts in RMS

Overall, these data fit into important general tumor dependency concepts. It has been recognized for a long time that tumor cells often not only depend on the driver oncogenes (oncogene addiction) but also on survival mechanisms pre-programmed in the cells of origin and involving master regulatory genes (lineage addiction) [115]. Acquired genetic alterations might affect and cooperate with these mechanisms, providing an explanation for lineage-restricted patterns of genetic alterations in cancer, which are prime characteristics for many sarcomas, including RMS. The concept of CRCs forms a framework that links oncogene- and lineage-addiction in both TF-driven PF+ and also PF− tumors. We argue that a deeper understanding of how acquired, RMS-relevant oncogenic alterations interact/cooperate with the cellular mechanisms that are pre-programmed in the cell-of-origins and/or define skeletal muscle identity will be key to breaking the code of RMS malignancy.

## 7. Epigenetic Regulators of Aberrant Myogenic Differentiation in RMS

During muscle differentiation, chromatin-modifying enzymes/remodeling complexes reprogram muscle promoters, alter chromatin structure through post-translational modifications and, ultimately, impact the activity of MRFs. A growing body of evidence has illuminated our understanding of the epigenetic mechanisms by which skeletal muscle differentiation is dysregulated in RMS.

### 7.1. Aberrant Epigenetic Control of MYOD1 Expression in RMS

Changes in gene expression during differentiation processes are orchestrated chiefly by changes in histone methylation and acetylation. Epigenetic regulation of MYOD1 activity by the SUV39H1 histone lysine methyltransferase (KMT1A/SUV39H1; hereafter KMT1A), histone deacetylase 3 (HDAC3), Lysine Acetyltransferase 2B (PCAF/KAT2B; hereafter PCAF) and Snail Family Transcriptional Repressor 2 (SNAI2) are examples of how different players coordinate to block MYOD1-dependent myogenic differentiation in RMS (refer to Figure 4 for a schematic representation of epigenetic enzymes involved in aberrant myogenic differentiation in RMS).

Firstly, KMT1A associates with MYOD1 and represses gene expression through trimethylation of lysine 9 on histone H3 (H3K9) [116,117]. In normal myoblasts, the MYOD1-dependent histone methyltransferase activity of KMT1A diminishes as differentiation proceeds due to the activation of p38α, which phosphorylates KMT1A and disrupts its association with MYOD1, thus enabling transactivation of downstream myogenic genes [118]. However, in P3F+ RMS cells, p38α signaling is defective [85]. In response to differentiation cues, histone methyltransferase activity of KMT1A and H3K9me3 levels increases and disrupts the transactivation of downstream myogenic genes by MYOD1 [85,117,118].

Secondly, HDAC3 is overexpressed in RMS tissue compared to normal muscle and emerged as the main HDAC restricting RMS differentiation from a high-efficiency clustered regularly interspaced short palindromic repeats (CRISPR)-based phenotypic screen of class I and II HDAC genes [119]. HDAC3 associates with nuclear receptor corepressor 1 (NCOR1) and nuclear receptor corepressor 2 (NCOR2) to form the NCOR/HDAC3 complex, which interacts with MYOD1 to suppress transcriptional activation of myogenic genes. Silencing of *HDAC3* enhances H3K9 acetylation in regulatory regions of MYOD1 target genes to induce differentiation [119].

Thirdly, recruitment of PCAF to the *MYF4* promoter in differentiating myoblasts acetylates *MYOD1* and thereby enhances its binding to DNA and ability to transactivate muscle genes [120,121]. Coactivator Associated Arginine Methyltransferase 1 (hereafter CARM1/PRMT4) associates with PCAF and facilitates its recruitment to the *MYF4* promoter [122]. In PF− RMS cells, PCAF is recruited to the *MYF4* promoter upon exposure to differentiation cues [123].

Finally, the MYOD1 differentiation axis is further regulated by the zinc finger TF SNAI2, which acts as a repressor of gene transcription and restricts differentiation in RMS. Downregulation of *SNAI2* in skeletal muscle precursor cells regulates the transition between proliferative cell states and differentiation [124]. In PF− RMS cells, overexpression of *SNAI2* is induced by MYOD1 through SE regulation [125]. SNAI2 competes with MYOD1 for binding to enhancer elements of muscle genes, and suppression of SNAI2 releases these enhancer regions and allows MYOD1 to activate the myogenic differentiation program and prevent tumor growth (refer to Figure 5 for the mechanism of action by which SNAI2 inhibits myogenic differentiation).

### 7.2. Chromatin Regulatory Complex PRC2

PRC2 works in concert with Polycomb repressive complex 1 (PRC1) to bind H3K27me3 marks and promote gene repression [126]. EZH2 is the catalytic subunit of PRC2 and trimethylates histone H3 at lysine 27 (H3K27me3). EZH2 plays a key role among epigenetic enzymes regulating tissue development. *EZH2* is downregulated during cellular differentiation and undetectable in normal adult tissues, but its expression and activity are deregulated in a variety of cancers [127]. In skeletal muscle, EZH2 repressive functions are required to delay differentiation and allow myogenic precursors to proliferate and fill the anatomic space in specific regions of the embryo.

In sarcomas, upregulation of PRC2 components—in particular *EZH2*—has been correlated with metastases and lower survival [128,129,130]. In PF− RMS, aberrant PRC2 activity—including aberrant EZH2 activity—has been shown to block skeletal muscle differentiation by repressing the transcription of myogenic genes (refer to Figure 4 for a schematic representation of epigenetic enzymes involved in aberrant myogenic differentiation in RMS) [131,132]. EZH2 is also recruited by YIN-YANG-1 (*YY1*) [133] and by GATA Binding Protein 4 (GATA-4) [134] to repress the expression of selected myogenic microRNAs in proliferating myoblasts and RMS cells.

Jumonji And AT-Rich Interaction Domain Containing 2 (JARID2), a Jumonji domain-containing interacting protein, use the PRC2 component Embryonic Ectoderm Development (EED) to regulate H3K27me3 in the promoter region of *MYF4* and Myosin Light Chain (*MYL1*). JARID2 is a direct transcriptional target of the fusion oncogene and uses EED to maintain an aberrant myogenic phenotype in P3F+ RMS [135]. Silencing of *JARID2* resulted in myogenic differentiation of P3F+ RMS [135].

### 7.3. Other Epigenetic Regulators of Aberrant Myogenic Differentiation in RMS

Euchromatic Histone Lysine Methyltransferase 2 (G9a/EHMT2; hereafter G9a) has been associated with tumor-promoting activity in several tumor types, including RMS [136,137,138,139,140,141]. G9a and EHMT1 are paralogs that perform the same enzymatic functions: catalyzing mono- or di-methylation of histone H3 at lysine 27 (H3K9me1/H3K9me2). G9a and EHMT1 cooperate by forming a heteromeric complex [142], but they are not completely functionally redundant and maintain their own unique roles [143]. In PF− RMS, G9a overexpression indirectly suppresses the canonical WNT signaling pathway to prevent myogenic differentiation by activating, rather than repressing, the expression of the WNT antagonist Dickkopf WNT Signaling Pathway Inhibitor 1 (*DKK1*) [141] through recruitment of Sp1 TF (SP1) and E1A Binding Protein P300 (P300) to the *DKK1* promoter. In PF+ RMS, G9a is regulated by the orphan receptor Nuclear Receptor Subfamily 4 Group A Member 1 (NR4A1, [109]), which drives the expression of the P3F fusion gene itself [144]. G9a directly represses Phosphatase and Tensin Homolog (PTEN) expression, thus increasing AKT Serine and Threonine Kinase (AKT) and RAC Family Small GTPase 1 (RAC1) activity and impeding Phosphoinositide-3-Kinase (PI3K) pathway activity [139].

Less is known about the oncogenic role of Euchromatic Histone Lysine Methyltransferase 1 (GLP/EHMT1; hereafter, EHMT1) in RMS. There is some evidence that silencing of *EHMT1* in P3F+ RMS cell lines results in decreased motility and induction of differentiation, along with reduced tumor progression in xenograft models in vivo [145]. Indeed, EHMT1 stabilizes the CCAAT-enhancer-binding protein (C/EBPb) to transactivate Aldehyde Dehydrogenase 1 Family Member A1 (*ALDH1A1*) and thereby maintain a stem-like state in RMS [146].

Finally, the TF SNAIL Family Transcription SNAI1 (SNAIL) [147] and the SWI/SNF Related, Matrix Associated, Actin Dependent Regulator Of Chromatin, Subfamily A, Member 4 (SMARCA4/BRG1) subunit of the mSWI/SNF chromatin remodeling complex [148] were identified as key suppressors of myogenic differentiation and promoters of oncogenesis in PF+ RMS.

## 8. MicroRNA (miRNA)-Dependent Post-Transcriptional Dysregulation of Myogenic Differentiation in RMS

### 8.1. MiRNAs in Cancer

Post-transcriptional regulation of gene expression by miRNAs, a class of small non-coding (nc) RNAs, is one of the key mechanisms allowing for a rapid adaptive response to environmental cues during development. MiRNAs are short ncRNAs of about 22 nucleotides, which are produced in the nucleus as pri-miRNAs and then processed by the Drosha complex into shorter precursors of about 70 nucleotides called pre-miRNAs. Pre-miRNAs translocate to the cytoplasm, where they are processed again by the DICER1 enzyme to give rise to mature miRNAs [149,150]. MiRNAs function by binding to complementary sequences on target mRNAs, resulting in translational repression or mRNA degradation, depending on the grade of complementarity between the seed region of the miRNA and the 3′ untranslated region (UTR) of the target mRNA [151]. MiRNAs always target multiple mRNAs, thereby mediating post-transcriptional control of several processes concurrently, including differentiation, proliferation and survival [149,152]. MiRNAs are crucial players in cancer malignancy and may behave as tumor suppressors or oncogenes depending on tumor context. Notably, germline variants in *DICER1* have long been recognized to confer genetic susceptibility to develop cancers, including RMS [80,82].

### 8.2. MYOmiR Family of miRNAs

MyomiRs are a group of miRNAs that are expressed in a tissue-specific manner in skeletal and/or cardiac muscle. During skeletal muscle differentiation, myomiRs are induced by MYF5, MYOD1 and MYF4 with the help of MEF2 factors through binding to muscle-specific intronic enhancers [153,154]. Specifically, the miR-1 family includes miR-1-1/miR-133a-2, miR-1-2/miR-133a-1 and miR-206/miR-133b bicistronic pairs, which are located on three different chromosomes [155]. MiR-1-1 and miR-1-2 are identical; only three nucleotides differ in miR-206. All carry the same seed sequence. Similarly, miR-133a-1 and miR-133a-2 are identical; only one nucleotide differs in miR-133b. Again, all three contain the same seed sequence [156]. Consequently, they share the same mRNA targets. MiR-206 is the only miRNA exclusively expressed in skeletal muscle [150]. MyomiR targets include Histone deacetylase 4 (*HDAC4*) [157] and the SWI/SNF Related, Matrix Associated, Actin Dependent Regulator Of Chromatin, Subfamily D, Member 1 (*SMARCD1*/*BAF60a*) subunit of the mSWI/SNF chromatin remodeling complex [158].

### 8.3. Deregulation of MYOmiRs in RMS

MiRNA deregulation contributes to RMS pathogenesis [150,159,160]. MiR-1 and miR-206 levels are lower in RMS compared to muscle [161,162,163], where low miR-206 levels correlated with poor survival, higher stage and presence of metastases in patients with RMS [161]. Low miR-206 levels in RMS appear to be under the control of SNAI1/2, and *SNAI1* silencing results in miR-206 up-regulation followed by cell differentiation [124,147]. Low miR-206 expression was linked to the expression of genes associated with muscle differentiation [161], and forced expression of pre-miR-206 in RMS cells halted proliferation, induced myogenic differentiation and inhibited tumor growth in vitro and in vivo.

The differentiating effects of MiR-206 in RMS were in part mediated by MET Proto-Oncogene, Receptor Tyrosine Kinase (MET) [163,164]. MiR-206 oligo miRNA mimics induced G0/G1 cell cycle blockade and inhibited cell migration in several RMS cell lines through suppression of *MET* expression [161]. We note that the proto-oncogene *MET*—a downstream target of both PAX3 and PAX3:FOXO1 [165,166]—plays an important role during skeletal muscle formation by favoring the delamination of myoblasts precursors and their migration to the limb bud [167,168]. MET also promotes migration and cell survival of RMS cells, and *MET* sequence variation, amplification and overexpression were observed in human RMS tumors [38].

Other MiR-206 targets in RMS include *PAX7* [147,169,170], NOTCH Receptor 3 (*NOTCH3*), and Cyclin D2 (*CCND2*) [163,164,170,171].

### 8.4. Deregulation of Other miRNAs in RMS

Certain miRNAs are not exclusively expressed in skeletal muscle but have pro-myogenic effects. In fact, the very first report on the involvement of miRNAs in the reluctance of RMS cells to differentiate pointed to the miR-29 family of miRNAs, which plays an important role in the NF-κB-*YY1*-miR-29 regulatory circuitry of myogenic differentiation. In brief, NF-κB activation sustains *YY1* levels and polycomb activity in myoblasts. *YY1* and EZH2 bind to a regulatory element upstream of miR-29b-2 and miR-29c on chromosome 1 to form a *YY1*/PRC2 repressor complex, which prevents the expression of miR-29a/b/c. Through a negative feedback loop, miR-29a/b/c precursors target the *YY1* 3′UTR mRNA region, reducing *YY1* protein levels and inducing differentiation [133]. Forced expression of miR-29 in RMS cell lines slowed down cell proliferation and inhibited tumor growth, increased Cyclin D1 (*CCND1*) and reduced *CDKN1A* expression, and induced cell cycle arrest and terminal muscle differentiation. These effects were mirrored by the silencing of *YY1*, which led to miR-29 induction, further supporting the important role of the *YY1*/miR-29 feedback loop in the governance of muscle differentiation in RMS [133,172].

A number of additional miRNAs have also been implicated in aberrant myogenic differentiation in RMS, including miR-214 (targeting *NRAS*) [173], miR-203 (targeting *TP63* and leukemia inhibitor factor receptor (*LIFR*)) [174], miR-378a-3p (targeting IGF1R) [175], and others. For example, miR-28-3p and miR-193a-5p are strongly upregulated in differentiating myoblasts, and their forced expression in RMS cells prevented invasion/migration, reduced proliferation, induced differentiation and inhibited tumor growth in vivo. Interestingly, overexpression of miR-193a-5p led to miR-206 upregulation, thereby supporting an intricate network of epigenetic miRNAs governing myogenesis [176,177].

### 8.5. Potential Avenues towards Therapeutic Targeting of miRNAs

Several RNA-based treatments (i.e., anti-sense nucleotides and small interfering RNAs) have gained FDA/EMA approval (e.g., Nusinersen, targeting Survival of Motor Neuron 2 (SMN2) pre-mRNA splicing in spinal muscular atrophy) or entered early phase clinical trials (e.g., for treatment of myeloid leukemias or carcinomas) [178]. In light of the clear role of miRNAs in dysregulated myogenic differentiation in RMS, a role for RNA-based treatments in RMS therapy may be possible.

## 9. Myogenic Differentiation as a Target for RMS Therapy

### 9.1. Differentiation Therapy in Cancer

Given the developmental nature of pediatric cancers, the idea of chemically inducing the differentiation of cells for therapeutic purposes—i.e., steering cancer cells into a benign direction—has long been entertained and pursued by pediatric oncologists [179]. Generally, terminal differentiation has been associated with growth arrest/blocking hyperproliferation and loss of migratory/invasive capacities of cancer cells. As a bonus, differentiating treatments often cause less therapy-related toxicity than conventional chemotherapy. In the clinical setting, two successful examples of differentiation therapies come to mind: The prime example is the treatment of acute promyelocytic leukemia (APL) with retinoic acid. APL used to be fatal, but it has become curable since two clinically effective therapies were identified—retinoic acid and arsenic—which trigger terminal differentiation of leukemic cells into granulocytes. A second example is the treatment of neuroblastoma, a neural crest-derived cancer. Because retinoic acid signaling partially controls differentiation of the neural crest into peripheral neural cells, *cis*-retinoic acid may be used to induce differentiation of residual neuroblastoma cells.

### 9.2. Overcoming the Differentiation Block in RMS as a Therapeutic Principle

As RMS cells are characterized by their reluctance to undergo terminal myogenic differentiation [180], there is significant interest in exploiting the causes of developmental stalling in RMS to restore differentiation as a therapeutic strategy. A notable early finding in RMS was that conventional chemotherapy increased the proportion of rhabdomyoblasts—variably differentiated, elongated, eosinophilic cells—which appeared to be further along the spectrum of maturation morphologically. Such cytodifferentiation, sometimes extensive, was more frequently observed in embryonal histology RMS, and it appeared to identify tumors that were responsive to therapy [181,182]. Extending beyond conventional therapy, pharmacological interventions that induce differentiation in RMS cells may provide means to overcome arrested myogenic development and/or render tumor cells more sensitive to previously established cytostatic agents.

### 9.3. Targeting Cell Cycle Progression in RMS

Integrated genomic, epigenomic and proteomic analyses of RMS cells derived from orthotopic PDXs highlighted activation of the cyclin D/CDK4/6-RB/E2F pathway and alteration of the G2/M mitotic spindle checkpoint pathway in RMS [89]. In fact, CDK4/6 inhibitor treatment of RMS cells caused G1 arrest accompanied by myogenic differentiation. This correlated with a reduction of tumor growth in vitro and in vivo [183]. However, palbociclib and abemaciclib—administered in combination with trametinib—failed to produce relevant anti-tumor effects in a randomized, preclinical phase II trial. By contrast, the WEE1 G2 checkpoint kinase (WEE1) inhibitor AZD1775—used in combination with vincristine and irinotecan—was found to be highly efficacious with a 70% complete remission/partial remission rate in a blinded, randomized, and placebo-controlled preclinical phase 3 trial [89]. WEE1 kinase is expressed highly in RMS [184] and regulates the G2/M checkpoint and DNA replication during the S phase of the cell cycle [185] by phosphorylating CDK1. WEE1 kinase inhibitor effects on myogenic differentiation have not been reported as of yet.

### 9.4. Targeting MAPK Signaling in PF− und PF+ Cells

In PF− RMS cell lines driven by *H/NRAS*-Q61X mutations, activation of the RAS pathway inhibits myogenic differentiation through the RAF-MEK-ERK effector pathway by repressing the expression of *MYF4* [186]. As a consequence, treatment with the Mitogen-Activated Protein Kinase Kinase 1/2 (MEK1/2) inhibitor trametinib releases transcriptional stalling at the *MYF4* promoter, followed by a widespread *MYF4*-induced change in SE landscape, myogenic differentiation and reduced tumor growth in mouse models [186]. Unfortunately, Yohe et al. demonstrated in preclinical models of PF− RMS that tumor cells rapidly acquired resistance to trametinib when used as a single agent [186].

Treatment with trametinib and IGFR1 inhibitors in combination could be a promising combinatorial approach; trametinib and the IGFR1 inhibitor ganitumab in combination were shown to be efficacious and well tolerated in preclinical models of RAS-mutant PF− RMS [187]. In an attempt to further optimize the therapeutic potential of trametinib, vertical double targeting of the RAF-MEK-ERK cascade, using RAF1 Proto-Oncogene, Serine/Threonine Kinase (RAF1) or pan-RAF inhibitors combined with MEK/ERK inhibitors, synergistically triggered myogenic differentiation and suppressed embryonal RMS tumor growth [188]. Currently, there are no specific RAF1 inhibitors available for clinical use, but several pan-RAF inhibitors are being studied in phase I/II clinical trials. Of particular interest is the pan-RAF inhibitor DAY101, which is currently being tested in a phase II clinical trial in pediatric low-grade glioma patients (NCT04775485).

### 9.5. Targeting Cellular Hierarchies in RMS

Recent scRNA sequencing efforts identified three developmentally distinct cell states (stem-like, myoblast/progenitor, and differentiated) in RMS. In PF− RMS, the mesoderm-like subset of cells was shown to be rather resistant to conventional chemotherapy and dependent on Epidermal Growth Factor Receptor (EGFR) signaling. Consequently, EGFR inhibitors may have a role in overcoming drug resistance of mesoderm-like cells within the PF− RMS cell pool. Indeed, EGFR inhibitors combined with irinotecan and vincristine significantly prolonged the survival of mice with embryonal histology PDX tumors [98].

For PF+ RMS, a subsequent pharmacological screen focused on measuring chemically induced changes in the abundance of the three main cell states and identified the MEK inhibitor trametinib as a strategy to rewire the cellular composition of PF+ RMS towards differentiated cell states [97]. When trametinib was combined with the multi-kinase inhibitors regorafenib, sorafenib or dabrafenib, the differentiation effect was increased, further supporting the notion that vertical double targeting of the MAPK pathway may be a strategy to interfere with RMS cell state transitions and overcome arrested differentiation in RMS. Co-treatment with trametinib and regorafenib (currently in clinical trials, NCT02085148) not only triggered differentiation but also suppressed PF+ RMS PDX tumor growth effectively [97].

### 9.6. Indirect Targeting of MYOD1 to Trigger RMS Differentiation

In large-scale CRISPR-based essentiality screens, *MYOD1* was among the most RMS-specific genetic vulnerabilities (DepMap; https://depmap.org/portal/ accessed 8 December 2022) [189]. Hence, interference with its activity by targeting key transcriptional regulators offers an RMS-specific therapeutic route. As described above and illustrated in Figure 4, SNAI2 is highly expressed and acts as a transcriptional repressor to inhibit MYOD1-driven differentiation and downregulate TFs necessary for terminal differentiation [125]. Incidentally, SNAI2 also represents an effector of the RAF-MEK-ERK signaling pathway. Trametinib blocks SNAI2, thereby allowing MYOD1 to bind to the enhancer elements of muscle-specific genes. Genetic removal of *SNAI2* induces myogenic differentiation and phenocopies the effects of MEK inhibition [125].

Other repressors of embryonal RMS differentiation were described. The TF *SIX1* maintains the cells in an undifferentiated state by reprogramming MYOD1 to occupy loci that drive tumor growth instead of muscle differentiation [190]. Loss of *SIX1* restores the *MYOD1*/*MYF4* gene regulatory network, induces tumor cell differentiation, and inhibits in vivo tumor growth [190]. Yet, TFs are challenging drug targets and indirect ways to interfere with their activity—e.g., by blocking the machinery associated with SE regulation—might be more straightforward.

### 9.7. HDAC and EZH2 Inhibitors

Interference with the activity of CRCs offers alternative therapeutic opportunities in RMS. High levels of histone modifications present at CRC-SEs make them vulnerable to inhibitors of epigenetic regulators [191], such as inhibitors of DNA demethylation and histone deacetylation.

In RMS, maintenance of the SE architecture appears to depend on appropriate histone acetylation [5,7]. Hyperacetylation due to HDAC inhibitor exposure spreads beyond the original SE borders and disrupts their 3D organization, thereby disrupting SE functionality [5]. HDAC inhibitors were shown to stimulate muscle differentiation and have been considered as a candidate drug in the treatment of muscular dystrophy [30]. Bharathy et al. demonstrated that the HDAC1/3 inhibitor Entinostat, administered in combination with vincristine, slowed PF− RMS tumor growth while inducing myo-differentiation in vivo [192], but pan-HDAC inhibitors have produced disappointing results. Phelps et al. suggested that the development of more selective HDAC inhibitors might improve anti-RMS efficacy [119].

The PRC2 component EZH2 has emerged as an important regulator of muscle differentiation and a candidate treatment target in RMS [128,129,130]. Various inhibitors of EZH2 have been used in preclinical studies. Treatment of RMS cells with MC1948 and MC1945—inhibitors of EZH2 catalytic activity—resulted in upregulation of *MYF4*, Creatine Kinase, M-type (*CKM*) and Myosin Heavy Chain (*MHC*) genes, and ultimately, skeletal muscle differentiation. These changes were reversed by enforced overexpression of *EZH2*, indicating an EZH2-specific effect [131,193,194]. The EZH2 inhibitor tazemetostat has been examined in a pediatric phase I trial (Appendix A; [195]) and may be considered in anti-RMS therapeutic strategies in the future.

Finally, a first-in-class dual EZH2/HDAC inhibitor 5 with (sub-) micromolar activity against both targets was introduced recently, representing another promising anti-RMS candidate compound [196].

### 9.8. Indirect Targeting of the PF Fusion Protein

In PF+ RMS, the differentiation block is mainly mediated by direct and/or indirect effects of the disease-driving fusion oncoprotein. As targeting of P3F remains challenging and possible plasticity in P3F expression [90,91] may counteract pharmacological interventions, substantial efforts have been directed towards identifying alternative options to alter PAX3:FOXO1 function. Key suppressors of myogenic differentiation and promoters of oncogenesis in PF+ RMS include transcriptional regulators such as JARID2 [135] and SNAIL [147], as well as the chromatin modifier SMARCA4 [148]. Pharmacological interference with their functions could overcome the differentiation block in PF+ RMS cells, but it has proven almost as difficult as targeting the fusion oncogene itself. Proteolysis targeting chimera degraders (such as PROTACs) may allow for the drugging of previously intractable therapeutic targets in the future by ubiquitinating and degrading them via the proteasome [197].

For the time being, tractable P3F/P7F targets such as Fibroblast Growth Factor Receptor 4 (FGFR4, [111,198,199]), Insulin-like Growth Factor Receptor 1 (IGF1R, [200]), and the MET Proto-Oncogene, Receptor Tyrosine Kinase receptor (MET, [165,166]) are thought to inhibit alveolar RMS differentiation and might offer windows of opportunity for (combination) therapy. Pharmacological inhibition of the histone methyltransferase G9a or the histone acetyltransferase PCAF would also be relatively simple. G9a inhibition prompted differentiation and decreased P3F+ RMS tumor growth in vivo as a result of increased *PTEN* expression [139]. PCAF overexpression in PF+ RMS was linked to higher affinity for *P3F* than for its normal target *MYOD1*, thereby acetylating and stabilizing the fusion protein. Inhibition of PCAF resulted in impaired PF+ RMS tumorigenesis in vitro and in vivo [192].

Epigenetic co-factors of P3F such as Bromodomain Containing 4 (BRD4), Chromodomain Helicase DNA Binding Protein 4 (CHD4), and Lysine Demethylase 4B (KDM4B) represent attractive targets for interference with oncogenic SE organization in PF+ RMS cells [113,201,202,203,204]. For example, P3F recruits BET bromodomain protein BRD4 [113] to establish extensive miswiring of enhancers and set up autoregulatory loops that maintain P3F expression [113], disrupt normal myogenic differentiation and lock RMS in an undifferentiated stage. Interference with BRD4 was shown to destabilize the expression of the fusion oncogene [113]. As BRD4 interacts with Cyclin-dependent Kinase 9 (CDK9) in other fusion-driven sarcomas, combinatorial treatment with BRD4 and CDK9 inhibitors may exert synergistic anti-tumor effects in PF+ RMS [205]. The combinatorial effects of BRD4 and CDK9 inhibitors on PF+ RMS cell cultures and PDX tumors are currently evaluated in a randomized, confirmatory preclinical study (animalstudyregistry.org; 10.17590/asr.0000286 accessed 18 March 2023).

### 9.9. Clinical Testing of Differentiating Agents in RMS

There are a number of chemical compounds that were investigated preclinically as inducers of differentiation in RMS and used in early-phase clinical trials for patients with pediatric cancer. Appendix A provides an overview of completed, terminated, active/non-recruiting and active/recruiting trials involving such compounds. Many trials did not recruit patients with RMS (e.g., NCT02601937 investigating the EZH2 inhibitor tazemetostat [195]). For HDAC inhibitors, no objective responses were observed in four patients with RMS and one patient with STS [206,207,208]. NCT02095132 investigated the effects of AZD1775 and irinotecan in patients with RMS (no response data available yet) but did not include vincristine despite preclinical studies clearly indicating that vincristine enhanced the anti-RMS efficacy of AZD1775 and irinotecan significantly.

Despite convincing data supporting the important role of MAPK signaling in muscle differentiation and RMS, we caution heavily against off-label trametinib monotherapy, as trametinib rapidly produced resistance in preclinical models of RMS [186]. Finally, we advocate very strongly for enrolling patients in early-phase clinical trials to systematically investigate the anti-RMS effects of candidate drugs!

## 10. Summary and Conclusions

RMS derives its name from the skeletal muscle features, which were observed microscopically in tumor tissue early on. Yet, more recent insights into the molecular mechanisms underpinning the disease have highlighted aberrant myogenic differentiation as a driving force in RMS malignancy.

This review summarizes our current understanding of the genetic and epigenetic characteristics of the disease as they determine aberrant muscle differentiation in RMS tissue. During embryonal development and regeneration of adult muscle, MRFs collaborate in directing skeletal muscle progenitor cells to establish and maintain the muscle lineage [4]. There is strong evidence to support the notion that, during the development and maintenance of RMS, MRFs collaborate with disease-driving oncogenic aberrations to initiate and sustain RMS cell proliferation, migration and aberrant differentiation [6,107,115]. Core features of RMS malignancy include deregulation of the epigenetically strictly controlled, hierarchically organized transcriptional events that define skeletal muscle development/differentiation and miswiring of muscle-specific CRCs.

RMS is a diverse cancer that manifests itself on a wide clinical and biological spectrum that goes far beyond the distinction of the four RMS histotypes recognized by the WHO [1,2]. In the clinical setting, adequate recognition of distinct RMS manifestations is a critical prerequisite for the adequate stratification of risk of treatment failure/death versus short-term/long-term treatment toxicity in a very young population of patients. Specific features of aberrant muscle differentiation observed in RMS tissue may aid in adequately recognizing distinct RMS manifestations [2] and likely result from the interaction of RMS-relevant driver oncogenes and mechanisms of survival/proliferation/differentiation pre-programmed in RMS cells-of-origin [115].

Differentiation of cells, including myogenic differentiation, is one domain in which cellular plasticity—i.e., the ability of cells to change their identity/states—is demonstrated. Plasticity may also serve as an escape mechanism enabling cells to adapt to changing environments. There is increasing evidence to support prominent plasticity within the RMS cell pool [90,91,97,98,99], especially with respect to PF+ RMS [90,91,97]. We argue that plasticity may be the mechanism at the root of PF+ RMS aggressiveness. Pharmacological interventions to overcome arrested myogenic development in RMS have garnered attention as a much-needed new window of opportunity.

Chemical compounds targeting muscle-relevant epigenetic pathways [119], the RAF-MEK-ERK cascade [97,186,188] and the stability of the fusion oncogene [113] have been brought forth as candidate anti-RMS drugs. Yet, several attractive treatment targets are not druggable thus far, and in the case of chemically tractable alterations, only a few compounds have entered clinical investigation due to limitations in drug specificity and/or selectivity. Future efforts should focus on rational combinations of drugs [97], dual-targeting of chemical compounds [209], and PROTACs to tackle previously undruggable targets by ubiquitinating and degrading them via the proteasome [197]. Ultimately, hope for the identification of new superior differentiation approaches in RMS and their use in rational combination therapies rests on future research deepening our understanding of RMS biology, including aberrant myogenic differentiation as a hallmark of this cancer.

## Figures and Tables

**Figure 1 cancers-15-02823-f001:**
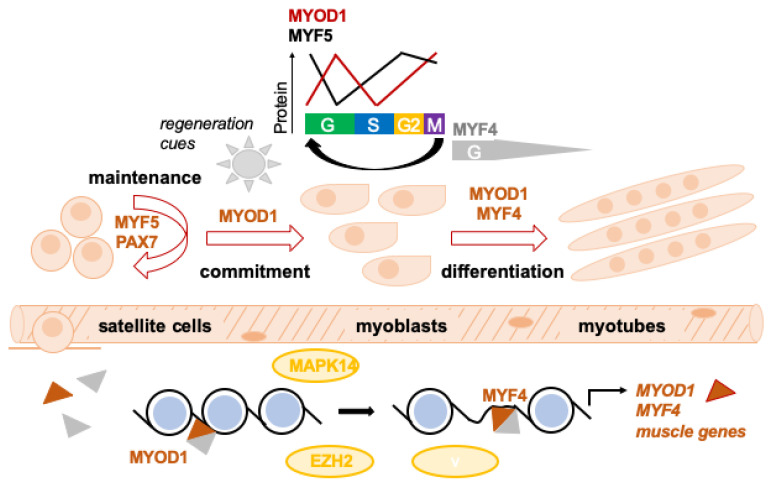
Normal skeletal muscle regeneration. Skeletal muscle is composed of bundles of terminally differentiated myofibers containing multiple post-mitotic nuclei. Muscle-resident stem cells, called satellite cells, are positioned beneath the basal lamina of mature myofibers. *MYF5* is the only active MRF expressed alongside *PAX7* in quiescent satellite cells in adult skeletal muscle. Upon muscle injury, MRFs are deployed to activate satellite cells, govern myoblast proliferation/fusion and, ultimately, withdrawal from the cell cycle and terminal differentiation. The transcriptional activity and protein availability of MRFs during myoblast proliferation/differentiation is tightly regulated. Regeneration cues rapidly induce *MYOD1* expression. Activated *MYOD1*-/*PAX7*-/*MYF5*-expressing satellite cells (myogenic precursors or myoblasts) continue to proliferate (with *MYOD1* and *MYF5* expression depending on cell cycle phase) and down-regulate *PAX7* while maintaining *MYOD1*, before they finally commit to myogenic differentiation via induction of *MYF4*. Other myoblasts maintain *PAX7* but down-regulate *MYOD1* and ultimately withdraw from the cell-cycle to regain quiescence and repopulate the satellite cell pool. This step-wise, hierarchical process is governed by complex epigenetic machinery. MRFs act as tissue-restricted transcription factors, which heterodimerize with bHLH proteins and bind to the regulatory regions of muscle-specific genes to activate transcription. MYOD1 binding also correlates with the opening of the chromatin structure at target genes through histone acetylation. Critical regulators of this process include p38α and a number of chromatin modifiers such as EZH2 and others.

**Figure 2 cancers-15-02823-f002:**
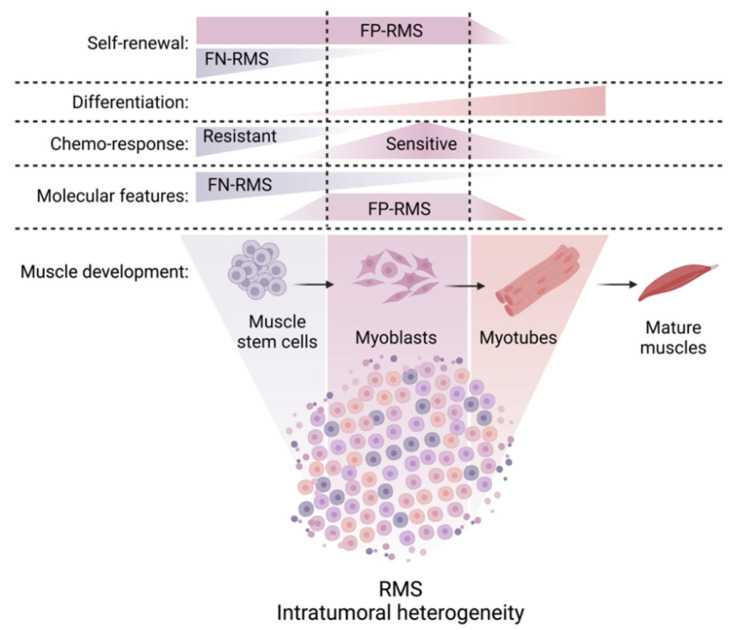
RMS cell-to-cell heterogeneity recapitulating distinct states of muscle development. The PAX::FOXO1 fusion-positive (FP) and PAX::FOXO1 fusion-negative (FN) RMS cell pool harbors cells stalled at distinct developmental states: a transcriptionally immature stem cell-like/mesoderm/mesenchymal state (expressing *PAX3*, *PAX7 CD44*), a myoblast state (expressing *MYF5*) containing highly proliferative/cycling cells, and a more differentiated myocyte state (expressing *MYF4*/*MYH3*/*MYH8*) associated with better patient outcomes. Conventional chemotherapy depletes the myoblast state and enriches cells in the mesenchymal state, which characterizes a tumor-propagating subpopulation capable of re-creating original tumor heterogeneity. The tumor cell pool in alveolar RMS is more skewed towards later stages of myogenesis prominent compared to embryonal RMS. FP RMS also displays more prominent intrinsic plasticity, which may be the mechanism at the root of its aggressive behavior.

**Figure 3 cancers-15-02823-f003:**
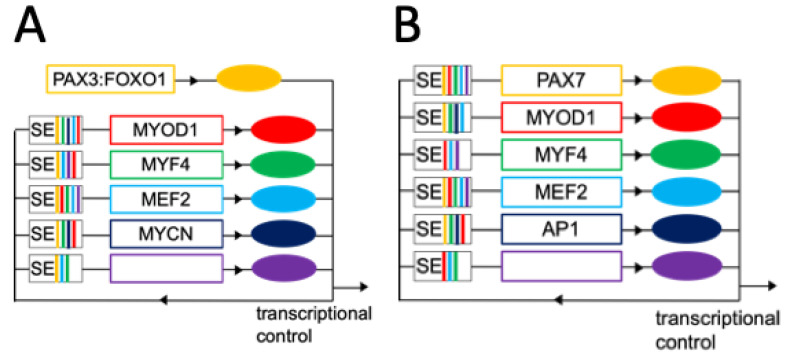
Core regulatory circuitries in RMS. Cell type-specific transcriptional programs are tightly controlled by master TFs (MTFs). CRCs are interconnected transcriptional loops in which individual MTFs control their own expression as well as the expression of the other involved TFs and exert feed-forward transcriptional control. These transcriptional networks control key transcriptional programs, which drive cell fate specification and differentiation. At the chromatin level, MTFs are accumulated at super-enhancers (SEs) and function under the control of SEs. CRCs have been identified for a number of different tissues, including normal skeletal muscle and RMS. Members of the MRF and MEF2 families are at the core of the normal muscle CRC, and MRFs appear to be the most common elements shared by CRCs in RMS. Generally speaking, the normal muscle CRC is active in RMS and complemented with additional modules: (**A**) The P3F+ RMS module relies on MYCN and P3F; the latter has pioneer factor activity and shapes the SE landscape in PF+ RMS cells. (**B**) The PF− RMS module involves *PAX7* and AP1 family TFs.

**Figure 4 cancers-15-02823-f004:**
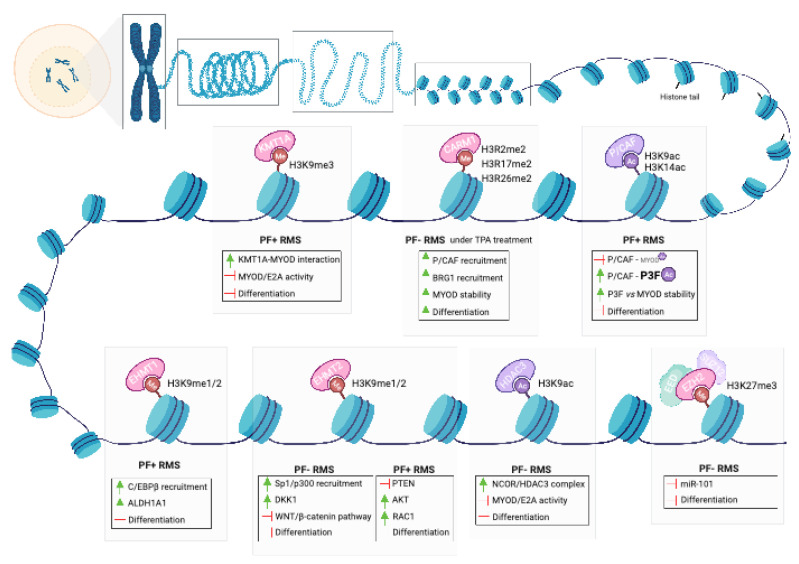
Epigenetic enzymes involved in aberrant myogenic differentiation in RMS. In the nucleus, human chromosomal DNA is organized in chromatin fiber loops, which are coiled around nucleosomes. The latter contain histones whose tails are subjected to post-translational modifications, including acetylation (Ac) and methylation (Me). This schematic representation summarizes the epigenetic enzymes, which have been reported to be responsible for histone modifications identified in RMS (green arrows indicate stimulation; red signs indicate blockade).

**Figure 5 cancers-15-02823-f005:**
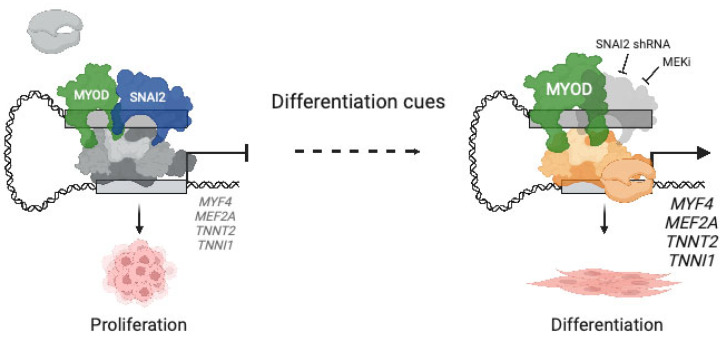
Inhibition of myogenic differentiation by SNAI2. In proliferating myoblasts and PF− RMS cells, SNAI2 binding at super-enhancers dampens MYOD1 activity at myogenic genes, supporting proliferation at the expense of differentiation (left panel). Upon receipt of differentiation cues, *SNAI2* is downregulated in differentiating myoblasts, thus permitting *MYOD1* transcriptional activity at myogenic differentiation genes (*MYF4*, *MEF2A*, *TNNT2*, *TNNI1*) and inducing muscle differentiation (right panel). In PF− RMS, the myogenic differentiation process can be achieved through *SNAI2* knockdown with short hairpin RNA (shRNA) alone or through treatment with the MEK inhibitor (MEKi) trametinib (right panel).

## Data Availability

Not applicable.

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
