# Peer review of "Genomic and Epigenetic Changes Drive Aberrant Skeletal Muscle Differentiation in Rhabdomyosarcoma"

_cancers, 2023, doi:10.3390/cancers15102823_

Round 1
Reviewer 1 Report
This work is an excellent, comprehensive review of the molecular changes that drive aberrant differentiation of skeletal muscle toward rhabdomyosarcoma (RMS). Aberrant myogenic differentiation, a driving force in RMS malignancy, has been identified through multiple molecular mechanisms. During embryonal development and adult muscle regeneration, myogenic regulatory factors (MRFs) direct skeletal muscle progenitor cells. In RMS, MRFs collaborate with oncogenic aberrations to initiate and sustain cell proliferation, migration, and aberrant differentiation. RMS is a diverse cancer that requires adequate recognition of distinct manifestations for proper risk stratification. There is evidence of prominent plasticity within the RMS cell pool, particularly with fusion-positive RMS, which may be the mechanism behind its aggressiveness. Pharmacological interventions that target muscle-relevant epigenetic pathways, the RAF-MEK-ERK cascade, and fusion oncogene stability have shown promise. However, several attractive treatment targets remain undruggable, and drug specificity and selectivity limitations have slowed clinical investigation. Future efforts should focus on rational drug combinations, dual-targeting of chemical compounds, and the use of PROTACs. A deeper understanding of aberrant myogenic differentiation in RMS is necessary to identify new superior differentiation approaches and develop rational combination therapies.
Minor changes:
· The body of the text appears monolithic in parts of the long essay. Further structuring with subtitles could improve the impression of comprehensibility of this long overview.
· Classification by methylation profiling feeds directly into considerations of the biological effect of silencing by methylation without differentiation. I would rather distinguish between these two approaches: Using methylation patterns for tumor classification and studying the biological effect of methylation of specific CpGs.
Author Response
We are grateful for the positive feedback and constructive criticism by the reviewer.
At the reviewer’s suggestion, the body of the text was restructured to improve the flow of information and comprehensibility. Also, we separated the review of methylation patterns for tumor classification and our comments on methylation of specific CpGs. We feel that these changes have substantially improved the manuscript.
Reviewer 2 Report
This is an interesting and well written review in the field. The subject is well exposed and up-to date and it will be of great interest for the readers of Cancers.
Author Response
We appraciet the reviewer's positive statement.